# From Biomarkers to Novel Therapeutic Approaches in Chronic Obstructive Pulmonary Disease

**DOI:** 10.3390/biomedicines9111638

**Published:** 2021-11-08

**Authors:** Hsu-Hui Wang, Shih-Lung Cheng

**Affiliations:** 1Department of Internal Medicine, Far Eastern Memorial Hospital, New Taipei City 10042, Taiwan; wangerge@gmail.com; 2Department of Chemical Engineering and Materials Science, Yuan-Ze University, Taoyuan City 320315, Taiwan

**Keywords:** chronic obstructive pulmonary disease, biomarkers, therapeutic approaches, biologics

## Abstract

Chronic obstructive pulmonary disease (COPD) is a heterogeneous and complex disorder. In this review, we provided a comprehensive overview of biomarkers involved in COPD, and potential novel biological therapies that may provide additional therapeutic options for COPD. The complex characteristics of COPD have made the recommendation of a generalized therapy challenging, suggesting that a tailored, personalized strategy may lead to better outcomes. Existing and unmet needs for COPD treatment support the continued development of biological therapies, including additional investigations into the potential clinical applications of this approach.

## 1. Introduction to COPD

Chronic obstructive pulmonary disease (COPD) is a primary contributor to chronic morbidity and mortality worldwide, and the population diagnosed with or dying from this disease is quite large, as COPD is the third leading cause of death and affects one-tenth of the global population [1]. The prevalence of COPD is expected to substantially increase over time, with an estimated five million annual deaths in 2006 due to COPD and its associated complications [2]. The complicated and heterogenous characteristics of COPD require the development of novel COPD therapeutic strategies that focus on lung function severity, symptoms, serum biomarkers, phenotypes, and comorbidities.

The factors that contribute to COPD development and the underlying mechanisms have been comprehensively studied. Although cigarette smoking is well recognized as a risk factor for developing COPD, no more than half of all smokers develop COPD during their lifetimes [3,4,5]. Early studies suggested that men were at greater risk of developing COPD than women, but recent pathological evidence suggests that women are more susceptible to COPD than men [6,7]. In addition to cigarette smoking, increased exposure to particulate matter (PM_2.5_ and PM_10_) due to severe urban air pollution, occupational exposures, and indoor air pollution due to biomass fuel combustion can also predispose individuals to the development of COPD [8,9,10,11]. People who suffer from chronic bronchitis have also been identified as being at higher risk of COPD [12].

## 2. Biomarkers for the Assessment of COPD

A growing body of research has focused on the use of biomarkers in COPD. Biomarkers are considered objective indicators that can be used to differentiate between normal and pathological status or determine the response and therapeutic efficacy of pharmacological treatments. However, the reliability of biomarkers for defining the disease state has been controversial due to poor associations with various disease phenotypes and poor reproducibility across patient cohorts [13,14,15]. Thus far, the assessment of eosinophils appears to be the only reliable marker for determining the potential efficacy of corticosteroids [14].

Under normal conditions, eosinophils comprise 1% to 4% of total leukocytes in the peripheral blood. The differentiation of a hematopoietic stem cell into a mature eosinophil is dictated by the presence of interleukin (IL)-5 [16]. Eosinophils increase substantially in number during type 2 helper T cell (Th2)-mediated inflammation and become essential effector cells during the inflammatory response [17,18]. Eosinophils circulate in the bloodstream and transmigrate to the bronchial vascular endothelium [16]. Inflammatory signals activate or induce the expression of adhesion molecules on both the bronchial vascular endothelium and epithelium, which allow for the eosinophil infiltration of the airway [19]. Eosinophils are attracted to inflammatory tissues by chemokines, such as C-C motif chemokine ligand (CCL)5, CCL7, CCL11, CCL13, CCL15, CCL24, and CCL26, and are activated by pro-inflammatory cytokines, such as IL-3, IL-5, and granulocyte–macrophage colony-stimulating factor (GM-CSF). Activated eosinophils in the airways release pro-inflammatory cytokines to maintain an inflammatory state, causing tissue damage [20]. Although Th-2-mediated eosinophilic airway inflammation is typical, neutrophilic inflammation is more frequently observed in the context of COPD, associated with Th1-mediated inflammation driven by the response of neutrophils to bacterial colonization [21]. Approximately 10–40% of patients with COPD present a degree of eosinophil-driven inflammation under stable conditions, and an eosinophil-predominant phenotype was identified in 28% of exacerbations [22].

Recent GOLD guidelines suggest that serum eosinophil populations can be used as a marker to identify phenotypes or predict ICS responsiveness [23]. Eosinophilic inflammation in COPD is known to be treatable in COPD. Several monoclonal antibody therapies against IL-5 (mepolizumab), IL-5 receptor-alpha (benralizumab), IL-13 (tralokinumab), and IL-4 receptor-alpha (dupilumab) have been developed to target inflammatory pathways [22,23,24,25]. Patients with higher levels of serum eosinophils during stable disease display an increased tendency to suffer from frequent episodes of severe exacerbation [26]. A multicenter, randomized control study was conducted to determine the recovery rates associated with eosinophilic and non-eosinophilic exacerbations among patients with COPD [27]. Exacerbation associated with serum eosinophil counts ≥200 cells/µL or ≥2% of the total leukocyte count were defined as eosinophilic exacerbations. The length of hospital stay following corticosteroid treatments was an average of 1.5 days shorter for eosinophilic exacerbations than for non-eosinophilic exacerbations (*p* = 0.015) [28]. These study results indicated that acute exacerbation associated with eosinophilic inflammation might benefit from fast response to corticosteroid, leading to a shorter hospital stay. Other cohort studies demonstrated that patients with low eosinophil counts (<50 cells/µL) were strongly associated with infection, resulting in a longer average hospital stay and a lower 12-month survival rate compared with patients with high eosinophil counts (>150 cells/µL) [29]. Thus, the serum eosinophil count might serve as a practical reference for deciding whether to administer corticosteroid therapy to patients with exacerbation.

Research regarding the reliability of serum eosinophil counts for the prediction of exacerbation occurrence or ICS treatment outcomes in patients with stable COPD remains unsettled. Although many studies have provided evidence linking eosinophilia with exacerbation risk and the potential for positive therapeutic effects due to ICS, eosinophils may not be a reliable reference for COPD diagnosis [30]. A systematic review of randomized controlled trials and observational studies performed post hoc analyses to examine serum eosinophil thresholds for ICS: relative eosinophil counts ≥2%, absolute eosinophil counts ≥150 cells/µL and ≥300 cells/µL [31]. A positive association with ICS response was demonstrated for eosinophils ≥2% and ≥150 cells/µL, but not for ≥300 cells/µL. No association between ICS and the risk of moderate or severe COPD exacerbation events was identified in the observational studies [32]. Using the CHAIN and BODE cohorts, serum eosinophil levels were measured at baseline and followed for two years to investigate the prevalence and stability of high serum eosinophil levels (≥300 cells/µL) and their relationships with the risks of future exacerbations. A significant proportion of patients with COPD in the study showed fluctuations in serum eosinophil levels, with only 12–15% of patients presenting with constantly high levels of blood eosinophils throughout the study period. No difference in the exacerbation rate was observed between patients with and without eosinophils [33].

Despite the uncertain role played by serum eosinophil levels for the prediction of COPD exacerbation events and ICS effects, a number of studies have presented evidence that blood eosinophil levels can be used to predict the ability of ICS to prevent exacerbations [34]. Patients with eosinophil counts >300 cells/µL were found to be the most likely to benefit from ICS treatment, whereas patients with eosinophil counts <100 cells/µL and >300 cells/µL were estimated to benefit from ICS treatment at various magnitudes [35]. Siddiqui et al. also demonstrated that LABA + ICS treatment achieved better therapeutic effects in patients with elevated blood eosinophil counts compared with ICS treatment alone, especially among patients with eosinophil counts ≥280 cells/µL [36]. By contrast, patients with eosinophil counts ≥2% who were treated with LABA + ICS therapy showed a significantly reduced exacerbation rate compared with placebo-treated patients, but no significant difference was observed when compared with mono-component–treated patients [37]. No significant differences in exacerbation rates were observed between the two treatment regimens for patients with relative eosinophil counts <2%. In another study, patients were grouped according to relative eosinophil counts, using a threshold of ≥2%, to evaluate the therapeutic effects of ICS (fluticasone propionate) [38]. ICS significantly reduced exacerbation rates in patients with eosinophil counts <2% compared with those treated with placebo, whereas no difference was observed for patients with eosinophil counts ≥2%. The discrepancies reported across different studies indicates that additional studies remain necessary to investigate the reliability of blood eosinophil counts as a biomarker for guiding ICS use.

Several studies have attempted to evaluate the practicality of using blood eosinophil counts to predict both treatment effects and exacerbation risks. A systematic review and meta-analysis that included five studies with a total of 124,976 patients with moderate to very severe COPD assessed the association between blood eosinophil count ≥2% and reductions in the exacerbation rate and pneumonia incidence following ICS treatment [38]. In this study, 60% of patients presented with serum eosinophil counts ≥2%. Within this patient group, patients treated with ICS showed a 17% reduction in exacerbation compared with patients without ICS treatment. However, the risk of pneumonia-related events was significantly elevated in patients with serum eosinophil counts ≥2% treated with ICS compared with those without ICS treatment. No significant difference was observed between with and without ICS treatment in patients with serum eosinophil counts <2%. Similar results were reported for a prospective observational study that aimed to evaluate the relationship between serum eosinophil counts and COPD exacerbation–related lung function loss [39]. Patients with serum eosinophil counts ≥ 350 cells/µL without ICS treatment experienced exacerbation events associated with a more rapid loss of lung function compared with patients who received ICS treatment. By contrast, Oshagbemi et al. reported no increase in the risk of all-cause mortality among patients who discontinued ICS treatment, regardless of the blood eosinophil counts [40].

In addition to blood eosinophil counts, other COPD biomarkers, such as serum IgE levels and fractional exhaled nitric oxide (FeNO), have been intensively investigated. Atopy refers to the hereditary tendency to generate IgE antibodies against common environmental allergens, such as proteins [41], and atopy is widely recognized to be an essential pathophysiological factor in the development of asthma. Although the contributions of atopy to the development of COPD have not yet been fully investigated, atopy is considered a COPD risk factor [42], and the positive rate of atopy in COPD has been reported between 15% and 40% [43,44]. Bożek et al. showed that 33.3% of patients with COPD had IgE-dependent sensitization to environmental allergens compared with only 11.5% of healthy individuals [44]. In the CODE cohort, allergic sensitization in patients with COPD was associated with an increased exacerbation rate and aggravated respiratory symptoms [45]. A 3-year follow-up study demonstrated that the presence of atopy (based on the detection of a specific IgE) was positively associated with cough and chest tightness [41].

FeNO has been identified as a useful biomarker for type 2 inflammation, and the Global Initiative for Asthma recommends the use of FeNO as a reference for clinical assessment and therapeutic guidance in patients with asthma [42]. FeNO can be reproducibly measured in a non-invasive manner [46]. However, the role of FeNO in COPD remains inconclusive. A systematic review and meta-analysis including 24 studies found that FeNO levels in patients with COPD were slightly elevated compared with those in healthy individuals [47]. No association was identified between FeNO levels and COPD exacerbation. Patients with COPD who were ex-smokers showed higher FeNO levels than current smokers. A scoping review also concluded that FeNO alone could not be used in clinical settings because no FeNO cutoff value has been established for use in therapeutic guidance for COPD treatment [48]. Yamaji et al. proposed two cutoff values for FeNO for predicting ICS responsiveness: 35 ppb was able to distinguish patients with a near-certain response, whereas 20 ppb was able to exclude patients with a near-certainty of no response [49]. However, the sample size of this study was small, and the patient groups were limited to ex-smokers; therefore, future studies remain necessary to validate the applicability of these cutoff values to all patients with COPD.

## 3. Novel Therapeutic Approach of COPD

Current pharmacological therapies for stable COPD include bronchodilators, such as β2 agonists, and anti-inflammatories, such as corticosteroids [50]. The optimal COPD treatment strategy is determined based on symptom assessments, such as the mMRC questionnaire, the CAT, and the Clinical COPD Questionnaire (CCQ), as well as predicted future risks of disease progression and exacerbations [51], with the aim of improving symptoms and reducing future risks. Effective treatments for COPD include long-acting bronchodilators, such as LAMAs and LABAs, which reduce hyperinflation, ease COPD symptoms, and decrease exacerbations [51]. Patients with COPD commonly present with type 1 inflammation characterized by a predominant proportion of macrophages and neutrophils and increased numbers of CD8^+^ and CD4^+^ T cells [52]. Some COPD patients present with eosinophilic inflammation and the clinical features of asthma, including reversible airway obstruction, enhanced airway hyper-responsiveness, and an improved response to corticosteroid therapy [53]. Biologics targeting eosinophilic inflammation have been successful for treating severe asthma, and several cytokines, including IL-4, IL-5, and IL-13, have demonstrated promising therapeutic effects in patients with COPD. An increasing number of studies have focused on targeting neutrophilic inflammation, and an overview of current biologics that are used for COPD treatment is presented in Table 1.

### 3.1. Targeting Eosinophilic, Type 2 Inflammation

#### 3.1.1. IL-5

IL-5 is among the best-studied cytokines involved in eosinophilic inflammation and is produced by CD4^+^ Th2 lymphocytes, innate lymphoid cells, and eosinophils. IL-5 differentiates eosinophils from precursors in the bone marrow and prolongs eosinophil survival in the airways. IL-5 and IL-5 receptors have been targeted for COPD treatment through the eosinophilic pathway [61,62]. Starting in 2000, a double-blind, randomized, placebo-controlled trial was conducted to investigate the therapeutic effects of a monoclonal antibody against IL-5 in asthma patients [63]. A single intravenous infusion of anti–IL-5 antibody resulted in a marked reduction in blood eosinophils for up to 16 weeks and a reduction in sputum eosinophils at four weeks. Anti–IL-5 (mepolizumab and reslizumab) antibodies reduce blood and sputum eosinophil counts, achieving 50% attenuation of bronchial submucosal eosinophils [64]. Additionally, a monoclonal antibody against IL-5 receptor (benralizumab) was able to induce antibody-mediated cell cytotoxicity, resulting in an even large reduction in eosinophil populations in the bronchial submucosa [64]. A randomized, double-blind, placebo-controlled, phase 2 study was performed in multiple countries to determine whether benralizumab reduced the occurrence of acute exacerbations in patients with COPD with eosinophilia [55]. Enrolled patients were randomly assigned to receive either benralizumab or placebo for a total of 12 weeks, but no significant difference in the annual rate of acute COPD exacerbations was observed between the two groups. In a post hoc analysis, patients with blood eosinophil counts over 250 cells/µL, or sputum eosinophil counts over 2%, showed better improvements in lung function and health status, suggesting a more prominent effect of the antibody treatment for the eosinophilic group. Two phase 3, randomized, placebo-controlled, double-blind, parallel-group trials (METREX and METREO trials) were conducted to evaluate the effects of add-on subcutaneous mepolizumab treatment in frequently exacerbating patients with COPD, including the assessment of efficacy and safety [53]. The results demonstrated that mepolizumab at a dose of 100 mg was associated with a lower annual rate of moderate or severe exacerbations compared with placebo among patients with COPD with an eosinophilic phenotype, defined as a peripheral blood differential eosinophil count of 2% or more, which equates to approximately 150 to 200 eosinophils per cube millimeter. All anti–IL-5 therapies are administered once per month; mepolizumab and beralizumab can be administered by subcutaneous injection, but reslizumab is administered intravenously [65]. The side effects of anti–IL-5 therapies include headache, nasopharyngitis, and local injection reactions [66,67].

Mepolizumab is the first biologic therapy that effectively reduced the occurrence of COPD exacerbations, with studies indicating that treatment strongly reduced the severity of eosinophilic inflammation. However, additional studies are necessary to examine long-term safety and the risks of increasing exacerbations.

#### 3.1.2. IL-4 and IL-13

IL-4 is essential for the differentiation of Th2 cells and, together with IL-13, enhances IgE secretion from B cells, increasing eosinophilic inflammation [68]. The preliminary results for IL-4 inhibitors failed to meet expectations, leading most studies to focus instead on blocking IL-13 or IL-4Rα, which can block both IL-4 and IL-13. Various approaches, including the use of decoy receptors to block IL-4Rα, have resulted in unsatisfactory outcomes due to poor target inhibition [69]. However, IL-13 increases the expression of inducible nitric oxide synthase by airway epithelial cells; therefore, FeNO has been evaluated as a biomarker for predicting the response to anti–IL-13 therapies. Treatment with lebrikizumab, an IL-13–specific blocking antibody, resulted in only slight improvements in FEV_1_, but no other improvements were observed for other symptoms, quality of life, or the occurrence of exacerbations [70]. The inconsistency and marginal effects of lebrikizumab were later confirmed by two phase III clinical trials, leading to the discontinuation of lebrikizumab for patients with COPD [71]. Another IL-13–blocking antibody, tralokinumab, demonstrated neither improvements in asthma symptoms nor reductions in the occurrence of exacerbations [72]. Dupilumab, a fully human anti–IL-4α receptor monoclonal antibody, inhibits IL-4 and IL-13 and appears to be effective in patients with moderate to severe asthma, regardless of baseline serum eosinophil counts [73]. All enrolled patients with uncontrolled persistent asthma were given dupilumab as add-on therapy every two weeks over a total of 24 weeks. Reductions in annualized exacerbation rates were observed for the overall population (70–75.5%), the subgroup with >300 eosinophils per µL (71.2–80.7%), and the subgroup with <300 eosinophils per µL (59.9–67.6%). No other promising results for anti–IL-4 and anti–IL-13 therapy have been reported in COPD.

#### 3.1.3. IL-33

IL-33 is predominately released from epithelial cells, alveolar type 2 epithelial cells, endothelial cells, mast cells, and fibroblasts [74,75]. The expression of IL-33 is stimulated by inhaled stimuli, such as allergens, infections, pollution, and cigarette smoke, leading to the activation, migration, and recruitment of innate and adaptive immune cells and the production of type 2 (Th2) cytokines, including IL-4, IL-5, and IL-13 [76,77]. In COPD, IL-33 signaling is complex. Increased IL-33 expression was found in whole lung samples obtained from patients with COPD, as well as in epithelial and endothelial cells [78,79]. IL-33 is also elevated in a caspase-4–dependent manner in peripheral blood mononuclear cells obtained from patients with COPD compared with those from healthy individuals, suggesting that IL-33 might also be an essential factor in COPD [80].

Several clinical trials have been conducted to determine the therapeutic effects of anti–IL-33 monoclonal antibodies for treating asthma and moderate to severe COPD [81]. Etokimab is an anti–IL-33 monoclonal antibody in a phase 2 proof-of-concept trial for the treatment of eosinophilic asthma. In this study, patients were given a single dose of etokimab, which resulted in improved FEV_1_ and reduced blood eosinophil counts. However, etokimab had no further therapeutic effects and failed to achieve significant differences compared with placebo on the primary endpoints for the treatment of patients with chronic rhinosinusitis and nasal polyps in clinical studies; thus, etokimab has been discontinued from further development.

A recent phase 2a trial was completed for the evaluation of itepekimab, an anti–IL-33 monoclonal antibody, in patients with moderate to severe COPD on a stable regimen of triple-inhaled or double-inhaled background maintenance therapy [66]. Enrolled patients aged between 40 and 75 years who were current or former smokers diagnosed with COPD were randomly allocated to receive 300 mg itepekimab or placebo every two weeks for 24 to 52 weeks. No significant difference in the annualized rate of acute COPD exacerbations was observed between the itepekimab and placebo groups, but a minor improvement was observed in FEV_1_ in the itepekimab group. When analyzing former smokers with COPD, itepekimab demonstrated a significant reduction in acute COPD exacerbations and improved FEV_1_ improvement compared with placebo. However, no therapeutic benefit of itepekimab was observed in current smokers. Two phase 3 trials are ongoing to further determine the efficacy and safety of itepekimab in former smokers with COPD.

### 3.2. Targeting Neutrophilic, Non-Type 2 Inflammation

#### 3.2.1. IL-17

IL-17 is produced by Th17 cells and has been described as an essential cytokine that mediates steroid-resistant neutrophilic airway diseases, such as severe asthma and COPD [82]. The rationale for evaluating IL-17 as a potential therapeutic target in COPD has been supported by in vitro data showing that IL-17 was able to induce cytokines, such as IL-6 and IL-8, for neutrophil recruitment and activation, and the inhibition of IL-17 using a monoclonal antibody resulted in a reduction in neutrophil numbers in the bronchoalveolar lavage fluid in an animal model [83]. A phase 2 clinical study was initiated to determine the efficacy and safety of the anti–IL-17 monoclonal antibody CNTO 6785 in patients with moderate to severe COPD [84]. Eligible patients were randomly assigned to CNTO 6785 and placebo treatment every two weeks over a total of 24 weeks. The trial did not meet any efficacy endpoints, and no significant differences in infection rates were observed between CNTO 6785 and placebo. A recent preclinical study demonstrated that the inhibition of IL-17, using either a monoclonal antibody or a small-molecule IL-17 blocker, effectively elevated glucocorticoid sensitivity in steroid-resistant neutrophilic airway inflammation [85]. This preclinical study suggested a novel mechanism for steroid resistance in type-17 neutrophilic airway inflammation, shedding light on other possible therapeutic strategies for COPD.

#### 3.2.2. TNF

Tumor necrosis factor (TNF) appears to play a role in the pathogenesis of COPD, and several mechanisms have been described regarding the contributions of TNF to COPD pathology [86]. TNF overexpression is observed in airways, amplifying neutrophilic inflammation. TNF induces apoptosis in normal cells, and TNF might act as a major factor in the cachexia that characterizes chronic inflammation and is a known comorbidity of COPD [87]. In addition, emphysema has been associated with apoptosis in the cells of the alveolar wall, suggesting a possible mechanism through which TNF might contribute to emphysema development [88,89]. A multicenter and double-blind study was conducted to evaluate the safety and efficacy of an anti-TNF antibody (infliximab) in patients with moderate to severe COPD. Enrolled patients with severe COPD were treated with either infliximab or placebo over 24 weeks; however, the results showed no treatment benefit for improving symptoms or lung function or reducing exacerbations.

#### 3.2.3. IL-1β

IL-1β is a primary cytokine that mediates the initiation and persistence of inflammation. A high level of IL-1 production has been reported in stable COPD, which increases during exacerbations [90]. Macrophages are the primary source of IL-1β, but other cell types can also secrete IL-1β, including neutrophils, fibroblasts, T cells, and bronchial and alveolar epithelial cells. Preclinical studies have indicated that IL-1β activity in the lung can induce phenotypes similar to those observed in COPD, including lung inflammation and emphysema [91]. Moreover, serum IL-1β levels were negatively correlated with FEV_1_ in patients with COPD [92]. Phase 1 and phase 2 studies have been conducted to evaluate the safety and efficacy of the anti-human IL-1β monoclonal antibody canakinumab in patients with COPD. Eligible patients with COPD were randomly assigned to receive canakinumab or placebo for a 45-week study period. The study results did not meet the primary endpoints, with no clear therapeutic effects on improvements in FEV_1_, FVC, SVC, or forced expiratory flow 25–75%. Whether canakinumab can serve as be an appropriate therapeutic option for COPD remains unclear.

#### 3.2.4. CXCR2

CXC chemokine receptor 2 (CXCR2) is activated by CXC chemokines, such as CXCL1 and CXCL5, both of which are elevated in sputum from patients with COPD and increase further during exacerbations [93]. Various small-molecule CXCR2 antagonists have been studied, demonstrating the ability to inhibit the neutrophil activation and migration associated with neutrophilic airway inflammation [94]. For example, an orally administered antagonist of human CXCR2, navarixin, was evaluated in a phase 2 proof-of-concept trial to determine safety and efficacy in the treatment of COPD [60]. Patients with COPD who were either non-smokers or current smokers were randomly assigned to receive either navarixin or placebo for a 6-month period. Patients who took daily 50 mg navarixin showed a reduced sputum neutrophil count and a minor improvement in FEV_1_, an effect that was more prominent in current smokers than former smokers [60].

The selective CXCR2 antagonist danirixin showed potent antagonism of CXCR2 activity in preclinical studies [95]. A recent phase 2b trial was performed to evaluate the safety and therapeutic efficacy of danirixin in patients with mild to moderate COPD [96]. Enrolled patients were randomly allocated to receive danirixin or placebo for a 6-month period. Treatment with danirixin did not demonstrate any meaningful clinical benefits in improving COPD symptoms or health-related quality of life, contrasting the results of earlier studies, indicating that patients with COPD might benefit from treatment with a CXCR2 antagonist. However, the patients with COPD who were treated with danirixin experienced more exacerbations, indicating the need for further evaluations of the safety and efficacy of the CXCR2 antagonists.

## 4. Conclusions

In this review, we highlighted recent updates in therapeutic modalities for COPD, focused on biomarkers involved in COPD and novel therapeutic approaches using biologic therapy. We expect that this review would inspire more investigations that can facilitate further individualized therapy for COPD. Adaptations of the materials included herein for educational and training purposes are also encouraged.

## Figures and Tables

**Table 1 biomedicines-09-01638-t001:** Biologics used to treat COPD.

Cytokine Target	Biologics	Delivery Route	Therapeutic Effects	Reference
IL-5	Mepolizumab	Subcutaneous injection every four weeks	Small reduction in exacerbations	[54]
IL-5Rα	Benralizumab	Subcutaneous injection every four weeks	Minor effects on FEV_1_	[55]
IL-33	Itepekimab	Two Subcutaneous injections every two weeks	Reduced exacerbation rates and improved lung function in former smokers with COPD	[56]
TNFα	Infliximab	Subcutaneous injection every four weeks	No effect	[57,58]
IL-1β	Canakinumab	Subcutaneous injection every eight weeks	No effect	[59]
CXCR2	Navarixin (CXCR2 antagonist)	Oral administration once per day	Minor effect on FEV_1_	[60]

## Data Availability

Not applicable.

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
