# Peer review of "From Biomarkers to Novel Therapeutic Approaches in Chronic Obstructive Pulmonary Disease"

_biomedicines, 2021, doi:10.3390/biomedicines9111638_

Round 1

Reviewer 1 Report

Dear reviewer:

Thank you for your comment and suggestion for our manuscript Biomedicine ID 1455391. Below letter is our response to your comment and change in revised manuscript.

Sincerely yours,

Hsu Hui Wang MD 

Author of biomedicines-1455391

Shih-Lung Cheng MD, PhD

Corresponding author of biomedicines-1455391

2021-11-04

Comment 1:

The paragraph explaining how emphysema contributes to airflow obstruction could be a little clearer. Emphysema is permanent enlargement of airspaces with destruction of alveolar walls and loss of supporting connective tissue and elastic tissue. Hogg Lancet 2004; 364: 709. Respirology 2014; 19: 970. These losses of supporting structures make the airways more collapsible during expiration. Over time this can produce hyperinflation. Hyperinflation is caused by two factors: 1. Reduced emptying of the lungs because of the airways being narrower in expiration than inspiration.  2. Reduced elastic recoil of the emphysematous lung relative to the expanding forces of the rib cage and so the chest tends to spring out. . ERJ 1998; 12: 248.This worsens during exercise due to reduced expiration time leading to dynamic hyperinflation.

Response 1:

We thank the reviewer’s comment. Since in this review article we focused on biomarkers involved in COPD and novel therapeutic approach using biological therapy and there has already other review articles focusing on pathophysiology of COPD, we decided to remove sections mentioning pathophysiology of COPD to make this review article more simplified.

Comment 2:

When describing symptoms of COPD, it might be useful to point out that some people with COPD do not report dyspnea because they become less and less active to avoid that uncomfortable. Thus, their primary symptom is inactivity. Physical activity is reduced in patients with COPD from GOLD stage II. Watz et al ERJ 2009; 33: 262. AJRCCM 2005; 171: 972. NEJM 1998; 338: 94. ERJ 2009; 33: 262. Resp Med 2010; 104: 1005. Nurs res 2001; 50: 195.

Response 2:

We thank the reviewer’s comment. Since in this review article we focused on biomarkers involved in COPD and novel therapeutic approach using biological therapy and there has already other review articles focusing on symptoms of COPD, we decided to remove sections mentioning symptoms of COPD to make this article more simplified.

Comment 3:

The authors are correct in pointing out that the normal cut off for FEV1/FVC varies with age. Resp Res 2012; 13: 13. For the sake of simplicity, clinical trials use a cut off of 0.7, but this will lead to over diagnosis in older patients and under diagnosis in younger patients. Resp Med 2007; 101: 2326. Thorax 2007; 62: 237.

Response 3:

We thank the reviewer’s comment. Since in this review article we focused on biomarkers involved in COPD and novel therapeutic approach using biological therapy and there has already other review articles focusing on diagnosis of COPD, therefore we decided to remove sections mentioning diagnosis of COPD.

Comment 4:

In section 3 “current treatment for COPD”, there is an exclusive focus on medicines used for COPD. Other important aspects in the management of COPD have been neglected, namely smoking cessation, regular exercise and pulmonary rehabilitation and vaccination against influenza, pneumococcus and SARS-CoV-2.

Response 4:

We thank the reviewer’s comment. Since in this review article we focused on biomarkers involved in COPD and novel therapeutic approach using biological therapy therefore we decided to remove sections mentioning current treatment of COPD to make this review article more simplified.

Comment 5:

The reviews of eosinophils and FeNO are clear and sensible. However, these sections, especially the about eosinophils, would kame more sense if they immediately preceded the sections on “novel therapeutic approach”. Could the eosinopihil data be mentioned when discussing the mepolizumab trials.

Response 5:

We thank the reviewer’s comment and add COPD with an eosinophilic phenotype, defined as a peripheral blood differential eosinophil count of 2% or more, which equates to approximately 150 to 200 eosinophils per cube millimeter in paragraph discussing the mepolizumab trials. We also removed other sections, only remained sections mentioning biomarkers involved and novel therapeutic approach using biological therapy to make this review article more simplified and accurate. 

Reviewer 2 Report

Comment 1:

the title of your review is too complex and not very accurate

the content does not adequately reflect it is rather a reiteration of the current guidelines but contains two interesting parts

1 that on serum biomarkers

2 that on interleukine-targeted therapies

 Therefore I strongly recommend you to combine the two parts in a paper bearing a title "Serum biomarkers and novel therapeutic approaches in COPD" or something of the kind

Response 1:

We thank the reviewer’s comment and change the title to ‘Serum biomarkers involved and novel therapeutic approaches in chronic obstructive pulmonary disease’ as the reviewer’s kind suggestion.

Round 2

Reviewer 2 Report

the draft is improved in value and readability